# PeerJ

# Captive rearing of the deep-sea coral *Eguchipsammia fistula* from the Red Sea demonstrates remarkable physiological plasticity

Anna Roik[1,3], Till Röthig[1,3], Cornelia Roder[1], Paul J. Müller[2] and Christian R. Voolstra[1]

[1] Red Sea Research Center, King Abdullah University of Science and Technology (KAUST), Thuwal, Saudi Arabia
[2] Coastal and Marine Resources Core Lab, King Abdullah University of Science and Technology (KAUST), Thuwal, Saudi Arabia
[3] These authors contributed equally to this work.

## ABSTRACT

The presence of the cosmopolitan deep-sea coral *Eguchipsammia fistula* has recently been documented in the Red Sea, occurring in warm ($>20\,°C$), oxygen- and nutrient-limited habitats. We collected colonies of this species from the central Red Sea that successfully resided in aquaria for more than one year. During this period the corals were exposed to increased oxygen levels and nutrition *ad libitum* unlike in their natural habitat. Specimens of long-term reared *E. fistula* colonies were incubated for 24 h and calcification (G) as well as respiration rates (R) were measured. In comparison to on-board measurements of G and R rates on freshly collected specimens, we found that G was increased while R was decreased. *E. fistula* shows extensive tissue growth and polyp proliferation in aquaculture and can be kept at conditions that notably differ from its natural habitat. Its ability to cope with rapid and prolonged changes in regard to prevailing environmental conditions indicates a wide physiological plasticity. This may explain in part the cosmopolitan distribution of this species and emphasizes its value as a deep-sea coral model to study mechanisms of acclimation and adaptation.

Corresponding author
Christian R. Voolstra,
christian.voolstra@kaust.edu.sa

## INTRODUCTION

The existence of ahermatypic and azooxanthellate coral species, so-called deep-sea corals, in the Red Sea had been anecdotally reported more than a century ago (*Marenzeller, 1906*). However, only until recently were the first live deep-sea coral specimens for biological measurements collected from the Red Sea (*Roder et al., 2013*). The distribution of deep-sea corals depends on various environmental factors such as temperature, oxygen levels, aragonite saturation, sedimentary regimes, substrate properties, currents, and food availability (*Roberts et al., 2009*; *Naumann, Orejas & Ferrier-Pagès, 2014*; *Gori et al., 2014b*).

An applicable temperature range for reef building azooxanthellate corals in a global habitat-suitability model was estimated between 5 and 10 °C (*Davies & Guinotte, 2011*). Until recently, scleractinian azooxanthellate corals in the Mediterranean deep-sea ($\leqslant$14 °C) were assumed to live at the upper limit of their thermal distribution range (*Orejas et al., 2011a*; *Naumann, Orejas & Ferrier-Pagès, 2013*). In contrast, the Red Sea provides a unique habitat with temperatures exceeding 20 °C throughout the water column. Additionally, low oxygen levels, a salinity level above 40 PSU, and little inorganic nutrients represent presumably challenging conditions for deep-sea corals (*Edwards & Head, 1987*; *Roder et al., 2013*; *Qurban et al., 2014*).

The coral *Eguchipsammia fistula* (*Cairns, 1994*) is known to occur in the Indo-Pacific, Australia, and New Zealand (*Van der Land, 2008*). In the Red Sea, *E. fistula* has been found at least three times: in the northern parts of the Red Sea *Marenzeller (1906)* found mostly dead or damaged coral colonies between 490 and 900 m, whereas *Qurban et al. (2014)* observed alive corals at depth of 683 m. In the central Red Sea, *Roder et al. (2013)* reported the discovery of live specimens between 230 and 320 m. *Roder et al. (2013)* and *Qurban et al. (2014)* described habitat conditions highlighting a warm, saline, oxygen- and nutrient-poor environment. *Roder et al. (2013)* furthermore collected corals for metabolic measurements.

In this study we provide for the first time information on successful long-term rearing (>1 year) of the azooxanthellate coral *E. fistula* collected from the Red Sea. To further understand the limitations that are posed by the distinct environmental constraints of the Red Sea and to assess the physiological plasticity of the deep-sea coral *E. fistula*, we collected and compared calcification (G) and respiration (R) rates from long-term reared corals, with on-board based measurements of specimens freshly collected from their natural habitat (*Roder et al., 2013*).

## MATERIAL AND METHODS

### Rearing conditions

*E. fistula* colonies were sampled from the central Red Sea (N22°17.831, E38°53.811 from 320 m) in May 2013 during leg 6 of the cruise 'KRSE 2013.' Sampling was conducted using a ROV equipped with a custom-made shovel-like basket (scoop). Once on-board the RV 'AEGAEO,' the corals were transferred into a closed aquaria system (six 110 L LDPE tanks for corals; one 240 L acrylic reservoir tank; Aqua Medic Ecorunner 3,700 for water flow) filled with deep-sea water and equipped with a chiller (Aqua Medic Titan 1,500; 21 ± 0.5 °C). Water exchange (deep-sea water, about 230 L per day) was ensured until the live corals were transferred to aquaria facilities at the King Abdullah University of Science and Technology (KAUST). Larger colonies were introduced unmodified into the rearing facilities. Smaller branches were either attached onto reef cement sockets (Reef Construct, Aqua Medic) or onto PVC plates with coral glue (Coral Construct, Aqua Medic). The rearing system consisted of a protein skimmer (Aqua Medic Turboflotor 5,000 baby ECO), filter (trickling towers, Aqua Medic), and a 250 L reservoir tank. The corals were distributed in four 125 L tanks (L-LDPE containers with PVC lids, Bürkle, Germany) each

equipped with submersible current pumps (Aqua Medic Nanoprop 5,000) that create a constant current in the tanks (flow rate $\sim$5,000 L h$^{-1}$). Coral tanks, reservoir, skimmer, and filter were connected to an open flow batch system with a controlled temperature of 21.3 $\pm$ 0.3 °C (chiller AquaEuroUSA MC-1/2 HP), which corresponds closely to the temperature of the corals' natural habitat.

Water changes in this closed 750 L system were performed at a rate of 300 L per week using unfiltered sea surface water collected from 14 km offshore of KAUST (approximately N22°15.1, E38°57.386). To reduce high salinity of surface water, reverse osmosis water (Milli-Q System, Millipore) was added to adjust the salinity to 40–41 PSU. Corals were fed twice a week *ad libitum* with *Mysis* (Crustacea, Eumalacostraca) and *Artemia* (Crustacea, Sarsostraca) (Ruto Red Label frozen fish food in blister packages). After feeding, leftovers and other particulate pollution (e.g., feces) were siphoned off from the tanks. Corals were reared in darkness; only during feeding and cleaning activities were corals sporadically exposed to light.

## Physiology

To assess the physiological performance in long-term reared corals, we measured respiration (R) and calcification (G) rates in five colonies. Prior to the experiment, feeding was suspended for 5 days. Similar sized (1–2 polyps) coral colonies were transferred without air exposure into custom-made acrylic 2.3 L chambers filled with seawater from the coral rearing setup. Each chamber was equipped with current pumps (flow rate $\sim$0.8 L min$^{-1}$). A control incubation was run in parallel containing a bare coral skeleton in order to correct for a possible chamber effect. All chambers were placed in a temperature-controlled water bath (chiller AquaEuroUSA MC-1/2 HP) and corals were incubated for 24 h. We measured oxygen concentration, temperature, and salinity at start and end of the incubation using a portable Multi-Parameter Meter (HACH HQ40d). We determined R and G as described in *Schneider & Erez (2006)*: we derived R from the depletion of dissolved oxygen over the 24 h incubation and subtracted variations in oxygen concentrations in the control chamber from the coral incubations.

G was determined applying the total alkalinity (TA) depletion method. Seawater samples (50 mL) were taken in duplicates before and after the incubation from each chamber using 0.45 μm syringe filters. Filtered TA samples were stored (1–2 h) at a stable temperature (21 °C) before automated acidimetric titration (Titrando 888, Metrohm AG, Switzerland). Correction of TA values for coral specimens was performed by subtracting control chamber values. Values were calculated as respiration and calcification rate per hour and normalized to coral tissue surface area. Live tissue cover was determined using the geometric approximation method (*Naumann et al., 2009*) and ranged between 45 and 91 cm$^2$.

Calcification and respiration of freshly collected *E. fistula* specimens were measured as described in *Roder et al. (2013)*: five replicates were incubated for 60–80 min in 1 L glass beakers on-board during the cruise 'KRSE 2011.' Further details are available in *Roder et al. (2013)*.

**Table 1 Environmental conditions of the azooxanthellate coral *E. fistula* in its natural Red Sea habitat and in captivity (KAUST aquaculture). Values are provided as means ± SD.**

| Environmental parameters | Natural habitat[a] | Aquaculture (KAUST 2013–2014) |
|---|---|---|
| Temperature [°C] | 21.5 | $21.3 \pm 0.3$[c] |
| Salinity [PSU] | 40.5 | $40.5 \pm 0.5$[c] |
| pH | 8.0–8.1 | $8.11 \pm 0.02$[b] |
| $O_2$ [mg L$^{-1}$] | 1–2 | $8.67 \pm 0.04$[c] |
| Total Alkalinity (TA) [μmol kg$^{-1}$] | 2,400–2,500 | $2,064 \pm 29$[b] |
| Aragonite saturation state $\Omega$ | 3.5 | $3.06 \pm 0.08$[b] |

**Notes.**

[a] Values from *Roder et al. (2013)*.

[b] Measured twice in triplicate in November 2013.

[c] Continuous measurements.

## Statistical analysis

All measurements in this study were determined as means ± SD. We compared the respiration and calcification rates between captive *E. fistula* (this study) and freshly collected specimens from *Roder et al. (2013)*. To assess differences between our 24 h incubation experiment of captive *E. fistula* ($n = 5$) and measurements from *Roder et al. (2013)* from 60 to 80 min incubations of freshly collected specimens ($n = 5$), the non-parametric Mann–Whitney $U$ test was applied using Statistica 10 (StatSoft Inc. 2011, version 10), as data were not normally distributed (Lilliefors test, $p < 0.05$). Results were regarded as statistically significant at $p < 0.05$.

## RESULTS AND DISCUSSION

### Rearing conditions

To assess physiology of the deep-sea coral *E. fistula* from the Red Sea, we established a basic aquaria-based rearing system (see Materials and Methods). Salinity, temperature, and pH were similar compared to the natural habitat, but oxygen, total alkalinity (TA), and aragonite saturation state differed from the corals' natural Red Sea habitat (Table 1). Lower TA and aragonite saturation state ($\Omega_{arg}$) values in the aquaculture were a result of added Milli-Q water to adjust surface water salinity to 40–41 PSU. Due to aeration of the seawater storage container, oxygen levels were increased in aquaculture. The increase in dissolved oxygen concentrations did not have any noticeable detrimental effects on *E. fistula*. Furthermore, reared coral specimens did not seem to be affected by changes of water density/pressure (between deep-sea and rearing conditions) contrary to other azooxanthellate corals (*Dullo, Flögel & Rüggeberg, 2008*; *Naumann, Orejas & Ferrier-Pagès, 2013*).

### Phenotypic differences

More than 90% of coral colonies survived (for more than one year) under the here-described rearing conditions. During this period, we observed notable tissue expansion and polyp proliferation. Captive corals displayed increased tissue thickness and tissue regrowth over former bare skeleton and even over substrate to which the corals were

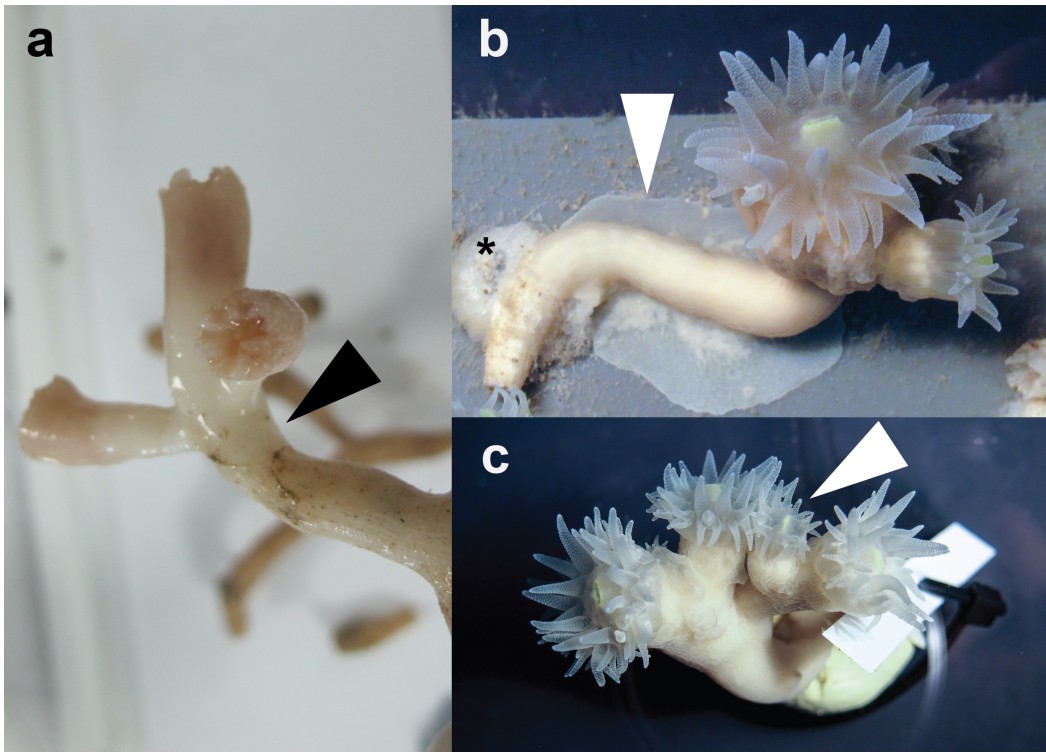

**Figure 1 Specimens of *E. fistula*.** (A) Freshly collected *Eguchipsammia fistula* specimen (arrowhead indicates tissue border). (B) Tissue overgrowing PVC substrate in long-term reared *E. fistula* (arrowhead indicates tissue; black asterix indicates aquaria glue). (C) Polyp proliferation in long-term reared *E. fistula* (arrowhead indicates newly grown polyp).

attached (Fig. 1B). Furthermore, larger sized polyps divided and/or new polyps budded at the distal end of long-term reared coral colonies (Fig. 1C). In contrast, freshly-collected *E. fistula* live tissue covered only the distal end (tip) of colony branches (Fig. 1A) (*Roder et al., 2013*).

## Respiration rates

Given our observation of tissue extension and polyp proliferation in aquaculture, we tested for an increase in metabolic activity. To do this, we compared respiration rates from incubations of long-term reared coral specimens to on-board collected measurements of *E. fistula* (*Roder et al., 2013*). Long-term reared corals were incubated under moderately different conditions than on-board measurements: Salinity, temperature, and oxygen rates were higher; pH, TA, and $\Omega_{arg}$ were lower (Table 2).

For the 24 h incubations of aquaria-reared corals, we found a decrease in $O_2$ concentration from 8.54 to 0.98 mg $L^{-1}$ over 24 h caused by respiratory processes. Oxygen levels from the control chamber indicated that a distinct portion of this respiration was not caused by the corals, where we measured a decrease from 8.63 to 2.22 mg $O_2$ $L^{-1}$ over 24 h. Therefore the control respiration was subtracted from the coral incubations. The resulting average respiration rate of *E. fistula* in aquaculture was 1.75 ± 0.78 µg $O_2$ $cm^{-2}$ $h^{-1}$. Interestingly, this is about 50% less compared to the average respiration rate of freshly

**Table 2 Incubation conditions (start values) and respiration (R) and calcification (G) rates of freshly collected (from on-board incubations) and long-term reared coral *E. fistula* specimens. Values are provided as means ± SD.**

| Parameter | On-board incubations (60–80 min) of freshly collected *E. fistula*[a] | Lab incubations (24 h) of long-term captive *E. fistula* |
|---|---|---|
| Temperature [°C] | $21.0 \pm 0.5$ | $21.8 \pm 0.1$ |
| Salinity [PSU] | 39.15 | $41.1 \pm 0.1$ |
| pH | 8.26 | $8.06 \pm 0.02$ |
| $O_2$ [mg L$^{-1}$] | $5.64 \pm 0.06$ | $8.54 \pm 0.05$ |
| Total Alkalinity (TA) [µmol kg$^{-1}$] | 2,433 | $1,842 \pm 15$ |
| Aragonite saturation state $\Omega$ | 5.45 | $2.50 \pm 0.07$ |
| R [µg $O_2$ cm$^{-2}$ h$^{-1}$] | $3.67 \pm 1.74$ | $1.75 \pm 0.78$ |
| G [µmol CaCO$_3$ cm$^{-2}$ h$^{-1}$] | $0.002 \pm 0.047$ | $0.013 \pm 0.01$ |

**Notes.**

[a] Values from *Roder et al. (2013)*.

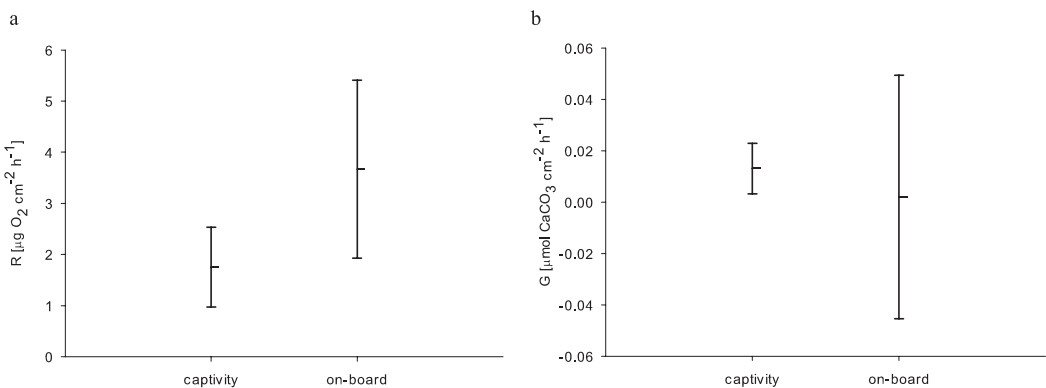

**Figure 2** (A) Respiration rates R and (B) Calcification rates G of long-term reared and freshly collected *E. fistula*; means ± SD.

collected specimens that were measured on-board (*Roder et al., 2013*) (Table 2). However, the difference in respiration rate was not statistically significant ($p_{U test} = 0.06$) (Fig. 2A).

*Naumann, Orejas & Ferrier-Pagès (2014)* found average respiration rates of $6.40 \pm 1.73$ µg $O_2$ cm$^{-2}$ h$^{-1}$ (corresponding to $4.8 \pm 1.3$ µmol $O_2$ cm$^{-2}$ d$^{-1}$) for *Lophelia pertusa* and $5.33 \pm 1.07$ µg $O_2$ cm$^{-2}$ h$^{-1}$ (corresponding to $4.0 \pm 0.8$ µmol $O_2$ cm$^{-2}$ h$^{-1}$) for *Madrepora oculata*. For the same two species, *Gori et al. (2014a)* found distinctive lower average respiration rates of $0.78 \pm 0.25$ µg $O_2$ cm$^{-2}$ h$^{-1}$ (corresponding to $0.47 \pm 0.15$ µmol C cm$^{-2}$ d$^{-1}$) and $1.07 \pm 0.23$ µg $O_2$ cm$^{-2}$ h$^{-1}$ (corresponding to $0.64 \pm 0.14$ µmol C cm$^{-2}$ d$^{-1}$), respectively. Furthermore, the authors reported average respiration rates of $2.23 \pm 0.52$ µg $O_2$ cm$^{-2}$ h$^{-1}$ (corresponding to $1.34 \pm 0.31$ µmol C cm$^{-2}$ d$^{-1}$) for *Desmophyllum dianthus* and $2.62 \pm 0.53$ µg $O_2$ cm$^{-2}$ h$^{-1}$ (corresponding to $1.57 \pm 0.32$ µmol C cm$^{-2}$ d$^{-1}$) for *Dendrophyllia cornigera*. In another study *Gori et al. (2014b)* found similar average maximum respiration rates for *D. cornigera* of $4.00 \pm 0.83$ µg $O_2$ cm$^{-2}$ h$^{-1}$ (corresponding to $2.4 \pm 0.5$

$\mu$mol C cm$^{-2}$ d$^{-1}$). One of the highest respiration rates, $5.00 \pm 1.33$ $\mu$g O$_2$ cm$^{-2}$ h$^{-1}$ (corresponding to $3.0 \pm 0.8$ $\mu$mol C cm$^{-2}$ d$^{-1}$), was reported by *Naumann et al. (2011)* from zooplankton-fed *D. dianthu*s specimens. In comparison, respiration rates of *E. fistula* are within the range of those determined for other azooxanthellate corals from colder habitats, but interestingly, at the lower end of the range, despite a distinctively higher water temperature ($>21$ °C).

## Calcification rates

In addition to respiration, we assessed calcification rates of reared *E. fistula* via the TA depletion method. The control chamber showed enriched TA values at the end of the incubation (after 24 h) with a difference of 16.36 $\mu$mol TA kg$^{-1}$. Thus, TA values were control corrected. The control corrected TA depletion method comprised values between 6.91 and 34.32 $\mu$mol TA kg$^{-1}$. We subsequently calculated an average calcification rate of $0.013 \pm 0.01$ $\mu$mol CaCO$_3$ cm$^{-2}$ h$^{-1}$ for long-term captive *E. fistula*.

Comparison of calcification rates from long-term captive *E. fistula* with results obtained from on-board incubations of freshly collected specimens by *Roder et al. (2013)* on average demonstrates a 6-fold increase of calcification rates for captive-reared corals (Fig. 2B). Yet due to a high variance associated with the on-board measurements, the difference was not statistically significant ($p_{U\,test} = 1.0$). In this study and in *Roder et al. (2013)* no correction for NH$_4^+$ effect on TA depletion method was performed, which has been shown to potentially underestimate calcification rates by up to 10 % (*Maier et al., 2012*).

Calcification in *D. cornigera* (an azooxanthellate species from the same family of *Dendrophyllidae*) ranged between 0.017 and 0.079 $\mu$mol CaCO$_3$ cm$^{-2}$ h$^{-1}$ at temperatures up to 16 °C (*Gori et al., 2014b*). Calcification rates for *E. fistula* compared to *D. cornigera* are similar despite higher water temperatures ($>21$ °C). It remains to be determined whether calcification of *E. fistula* from cold water environments and the Red Sea display similar rates at their corresponding prevailing habitat temperature, and accordingly if a relative, not an absolute increase in temperatures enhances calcification as has been shown for other corals (*Naumann, Orejas & Ferrier-Pagès, 2013*; *Naumann, Orejas & Ferrier-Pagès, 2014*).

## Effect of diet

It has been argued previously that reduced tissue cover and growth rates might represent coral adaptations to 'deep-sea conditions' in the Red Sea (*Roder et al., 2013*). Although there is no information available on the natural diet of *E. fistula* in the Red Sea, food sources for other azooxanthellate corals have been described as phytodetritus, phytoplankton, (zoo)plankton, and dissolved organic matter (*Gori et al., 2014a*). As observed for captive *E. fistula*, regular feeding and uptake of crustaceans is in line with the earlier reported hypothesis of other deep-sea corals capturing and utilizing (zoo)plankton (*Dodds et al., 2009*; *Qurban et al., 2014*). An increased food supply in captivity can result in increased growth as reported for other azooxanthellate corals (*Orejas et al., 2011b*), or conversely decreased growth when feeding is omitted (*Naumann et al., 2011*). Our results confirm that regular feeding and elevated oxygen levels (among others) induced increased tissue extension, polyp proliferation, and calcification rates (by trend) in *E. fistula* (Figs. 1 and 2).

### Experimental variables

Considering increased growth, food uptake, and available oxygen, the here-measured decrease (by trend) in respiration of captive-reared corals in comparison to freshly collected specimens seems less plausible. However, higher respiration rates in the on-board measurements might be interpreted as a response to stress (*Dodds et al., 2007*), possibly caused by the sampling procedure and rapidly changing environment from the depth to on-board incubation beakers (*Hennige et al., 2014*). These effects may also explain the higher variation in calcification of freshly collected corals (*Roder et al., 2013*) compared to long-term reared *E. fistula* (Fig. 2B).

## CONCLUSION

Our results show successful transferal of *E. fistula* coral specimens from nutrient- and oxygen-limited deep-sea habitats of the Red Sea into a rearing system that only in part corresponds to its natural environment. In contrast to other azooxantheallate coral species (*Naumann et al., 2011*), *E. fistula* is easy to maintain in a basic aquaria system demonstrating a wide physiological plasticity. The ability to cope with a wide range of physico-chemical conditions might be one of the clues to the cosmopolitan success of this species, as well as to the feasibility of aquaria-rearing in markedly different conditions compared to the species' natural habitat. Comparative experiments with *E. fistula* from different environments (e.g., Red Sea, Indo-Pacific, Australia, and New Zealand) in connection with population genetics studies will further reveal the extent of plasticity and connectivity to elucidate the ability of this widespread species to cope with environmental change.

## ACKNOWLEDGEMENTS

We thank the crew of the RV 'AEGAEO,' especially the ROV team for support in coral specimens collection. We thank AM Al-Suwailem and CMOR for assistance in coral rearing and cruise preparation.

### Funding

Research reported in this publication was supported by Center Competitive Funding (CCF) program award URF/1/1973-02 and baseline funds to CRV by King Abdullah University of Science and Technology (KAUST). The funders had no role in study design, data collection and analysis, decision to publish, or preparation of the manuscript.

### Grant Disclosures

The following grant information was disclosed by the authors:
Center Competitive Funding: URF/1/1973-02.
Baseline funds.

### Competing Interests

The authors declare there are no competing interests.

## Author Contributions

- Anna Roik and Till Röthig conceived and designed the experiments, performed the experiments, analyzed the data, wrote the paper, prepared figures and/or tables, reviewed drafts of the paper.
- Cornelia Roder analyzed the data, reviewed drafts of the paper.
- Paul J. Müller performed the experiments, contributed reagents/materials/analysis tools, reviewed drafts of the paper.
- Christian R. Voolstra conceived and designed the experiments, analyzed the data, contributed reagents/materials/analysis tools, wrote the paper, prepared figures and/or tables, reviewed drafts of the paper.

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
