# Peer review of "Captive rearing of the deep-sea coral Eguchipsammia fistula from the Red Sea demonstrates remarkable physiological plasticity"

_PeerJ, doi:10.7717/peerj.734_

## Round 0.1 · original submission · Major Revisions

As you can see, one of the reviewers has raised substantial concerns about this ms, which I concur. It is very clear that the ms will be substantially improved if you can appropriately address these questions, particularly regarding the details of your M and M section.

·

Basic reporting

Dear Authors,

As a Cold-water coral researcher I consider your study interesting, especially because I have been developing quite a bit research in aquaria developing ecophysiological experiments.
The coral protagonist of your experiment seems to be really interesting and I like the idea of the paper.
Unfortunately I have several concerns that in most cases can be resolve dedicating some more time to the manuscript.
My general opinion is that this manuscript is potentially interesting, as the authors also mentioned in their brief, for many coral researchers (from the Red Sea, Atlantic, Mediterranean and elsewhere) but to be really interesting, this paper needs to be presented in a much wider context. For this reason needs I consider that this paper deserve a major revision and I strongly recommended a second review process before the manuscript can be accept for publication.
In the next paragraphs I will make several comments and suggestions with the hope that they will be useful for the authors to improve and enlarge the manuscript.

No comments on the title and key words


Introduction

As mentioned in my general comments, this work is interesting, but reading the introduction it seems there is very few previous works done in ecophysiology of CWC. This is probably true for the red Sea, but not for other areas, and indeed there are already several published papers dealing with the same questions you have in your work: the degree of plasticity of CWC in front of environmental changes. This is an important question and indeed you can also put your study in this frame, as more knowledge is needed to better understand the response of different organisms (especially the ones which have a calcium carbonate skeleton) in front of climate change.
In the last 4-5 years several studies dealing with these topics have been published. Some of them with Cold-water coral Mediterranean species as target organisms: Naumann et al. 2011, 2013,2014; Gori et al. 2011, 2013, 2014a,b; Orejas et al. 2008, 2011a,b, Movilla et al. 2014a,b; but there is also quite a lot of work developed in Atlantic waters: Dodds et al. 2007, 2009, Hennige et al. 2014 among others. Please develop a wide comprehensive introduction putting your subject in context, this will make the paper much more interesting for more researchers.
Please pay special attention to the recent work of Gori et al. 2014b: Physiological performance of the cold-water coral Dendrophyllia cornigera reveals its preference for temperate environments.

Material and Methods
This is to my opinion one of the weakest parts of the manuscript as very few is explained about the
I miss a clear description of the experimental set up. Even if it is clear to me that these were two different experiments, it is important to have a clear picture on the two different experimental situations, not only related to the water parameters, but also to the kind of tanks, volume, current speed in the tanks, conditions, treatments conducted, number of exemplars etc.
Please explain a bit more about the used methods, even if you applied known techniques it will be good to know (in order to be able to see how comparable are your results with previous published ones) the procedure you follow. How did you measure Oxygen consumption? Please explain widely the used methods.

Results and discussion
More information on the methods will help to better understand the results you obtained and (1) see how comparable are the results from your two different start points: experiments on board and in the lab, (2) how comparable are your results with previous results from other researchers for other CWC species.
As mentioned in my comments to the introduction, there are many works already published dealing with the topic you present here and to know how singular that you found is, it is necessary to take into account all these previous studies, even if they have not been performed in the Red Sea. Further the consequences of these results in the future scenarios predicted for instance by the IPCC, are also important aspects to consider here. You discuss your results considering only two previous publications, which is not that much. Some data analyses are also desirable to present your results in a much more robust and quantitative way, otherwise they remain very descriptive and as far as I can see you have data to perform some analyses.

Experimental design

(Already included in the basic reporting)
Material and Methods
This is to my opinion one of the weakest parts of the manuscript as very few is explained about the
I miss a clear description of the experimental set up. Even if it is clear to me that these were two different experiments, it is important to have a clear picture on the two different experimental situations, not only related to the water parameters, but also to the kind of tanks, volume, current speed in the tanks, conditions, treatments conducted, number of exemplars etc.
Please explain a bit more about the used methods, even if you applied known techniques it will be good to know (in order to be able to see how comparable are your results with previous published ones) the procedure you follow. How did you measure Oxygen consumption? Please explain widely the used methods.

Validity of the findings

(Already included in the basic reporting)
More information on the methods will help to better understand the results you obtained and (1) see how comparable are the results from your two different start points: experiments on board and in the lab, (2) how comparable are your results with previous results from other researchers for other CWC species.
No data analyses has been made, this is also needed.

Additional comments

(all comments already included in the basic report)

Reviewer 2 ·

Basic reporting

This manuscript is a nice and timely contribution to the field of experimental ecology on deep-sea corals. It is well written and - beginning from the abstract - a neat story without any focus on spectacular yet speculative findings. Some (technical) aspects should be clarified/complemented in order to increase its level of accurateness. Therefore, my recommendation is to accept for publication after minor revisions (see next section).

Experimental design

In general, the methods and applied techniques are clearly described and easy to follow. However, I was wondering why you don't performed any statistical analysis? I therefore strongly recommend to compare/test your R and G data with the data from Roder et al. 2013 for statistical significance to underpin your statements.

Some specific questions:
- L48: How was sampling conducted (dredge/ROV/manned submersible)? Please give a short explanation or reference (Roder et al. 2013?)
- Rearing conditions: what about dissolved nutrients? In a closed recirculating system I assume a permanently increase of at least phosphate and nitrate (if your biofilter works well). Have you measured any nutrients? If yes, I would like to encourage you to present the data together with all other routinely measured parameters (temp, salinity, pH, etc.). I guess this would significantly improve the expressiveness of your long-term cultivation.
- L60: "corals were fed twice a week ad libitum"; are you confident that food supply was representative of what the corals are experiencing in the field? Please provide the details (e.g. energetic considerations based on respiration rates...).
- Respiration measurements: what were the final oxygen concentrations after the 24hrs in the respiration chambers? It would be crucial to know, if oxygen concentration after the incubations was on a normoxic range or if oxygen depletion was a stressor itself.
In this context: how were the data of Table 2 (Lab incubations) assessed? Are these initial or end values of the water properties?

Validity of the findings

no objections

Additional comments

no further comments

---

## Round 0.2 · accepted · Accept

As we have problems in identifying suitable reviewers, the review process has been unnecessarily long. Sincere apology.

After going through your revision and reviewers' comments received earlier on, I decide to accept your ms as it is for publication.

Happy New Year.